# Single molecule poly(A) tail-seq shows LARP4 opposes deadenylation throughout mRNA lifespan with most impact on short tails

Sandy Mattijssen[1], James R Iben[1], Tianwei Li[1], Steven L Coon[1], Richard J Maraia[1,2]*

[1]Intramural Research Program, *Eunice Kennedy Shriver* National Institute of Child Health and Human Development, National Institutes of Health, Bethesda, United States; [2]Commissioned Corps, U.S. Public Health Service, Rockville, United States

**Abstract** La-related protein 4 (LARP4) directly binds both poly(A) and poly(A)-binding protein (PABP). LARP4 was shown to promote poly(A) tail (PAT) lengthening and stabilization of individual mRNAs presumably by protection from deadenylation (Mattijssen et al., 2017). We developed a nucleotide resolution transcriptome-wide, single molecule SM-PAT-seq method. This revealed LARP4 effects on a wide range of PAT lengths for human mRNAs and mouse mRNAs from LARP4 knockout (KO) and control cells. LARP4 effects are clear on long PAT mRNAs but become more prominent at 30–75 nucleotides. We also analyzed time courses of PAT decay transcriptome-wide and for ~200 immune response mRNAs. This demonstrated accelerated deadenylation in KO cells on PATs < 75 nucleotides and phasing consistent with greater PABP dissociation in the absence of LARP4. Thus, LARP4 shapes PAT profiles throughout mRNA lifespan with impact on mRNA decay at short lengths known to sensitize PABP dissociation in response to deadenylation machinery.

**\*For correspondence:**
maraiar@dir6.nichd.nih.gov

**Competing interests:** The authors declare that no competing interests exist.

## Introduction

Critical aspects of mRNA metabolism are controlled by the 3' poly(A) tail (PAT) and the cytoplasmic poly(A) binding protein (PABP, PABPC1) which contribute to mRNA function (*Mangus et al., 2003*; *Nicholson and Pasquinelli, 2019*; *Thompson and Gilbert, 2017*). PABP is a translation factor which by binding to eIF4G can link the mRNA 5' cap and associated initiation factors with translation termination factors and the 3' PAT (*Uchida et al., 2002*). Beyond translation, PABP is a key factor involved in control of PAT length and mRNA stability. Nascent precursor-mRNAs acquire PATs in the nucleus and enter the cytoplasm with lengths of ~250 nt that shorten thereafter with gene-specific multifaceted kinetics (reviewed in *Chen and Shyu, 2017*; *Jalkanen et al., 2014*; *Nicholson and Pasquinelli, 2019*; *Yamashita et al., 2005*).

Multiple PABP molecules can bind to poly(A) with a periodicity of ~27 nt (*Baer and Kornberg, 1980*; *Nicholson and Pasquinelli, 2019*). PABP architecture is conserved, consisting of four RNA recognition motifs (RRMs), followed by a helix, a flexible linker (*Schäfer et al., 2019*) and a globular C-terminal Mademoiselle (MLLE) domain with a binding site for a PAM2 peptide. About 20 distinct proteins contain PAM2 peptides, of which several play key roles in mRNA metabolism and/or translation (*Xie et al., 2014*).

While earlier experiments had shown PAT shortening after PABP depletion from cell extracts, PATs became longer after PABP depletion in vivo (*Sachs and Davis, 1989*). PABP promotes PAT shortening by recruiting the PAM2-containing proteins, PAN3, Tob1, Tob2, GW182 and their associated deadenylases (reviewed in *Xie et al., 2014*). The PAN2–PAN3 and CCR4–NOT (CNOT)

complexes employ multiple deadenylases that can act during different phases of PAT shortening and are activated or blocked by PABP in ways that yield complex mRNA decay patterns reflective of varying deadenylation rates (*Chen et al., 2017*; *Webster et al., 2018*; *Yamashita et al., 2005*; *Yi et al., 2018*). In addition, deadenylases can be actively recruited in response to different cues, for example, involving AU-rich element binding proteins or miRNA-mediated mRNA decay factors (*Fabian et al., 2009*; *Huntzinger et al., 2013*). Thus, deadenylation is a major target for mRNA regulation (*Chen and Shyu, 2017*; *Chen and Shyu, 2011*), and several of the activities involved noted above are integrated by PABP including by the PAM2 system.

LARP4, which is also known as LARP4A directly binds poly(A) RNA and PABP (*Cruz-Gallardo et al., 2019*; *Yang et al., 2011*). LARP4 is a short-lived protein whose accumulation levels can span a wide range due in part to control by independent instability elements located in the coding region and in the 3'UTR of its mRNA (*Mattijssen et al., 2017*; *Mattijssen and Maraia, 2016*). As with other La-related proteins (LARPs), LARP4 contains a La motif followed by a RNA recognition motif (RRM), and LARP-specific domains (*Maraia et al., 2017*). LARP4 binds PABP via two motifs, an N-terminal variant PAM2 named PAM2w and a downstream PABP-binding motif (PBM), both of which are required for efficient polysome association and stabilization of poly(A)$^+$ mRNA (*Mattijssen et al., 2017*; *Yang et al., 2011*).

LARP4 effects on PATs were documented as their apparent net-lengthening on reporter as well as endogenous mRNAs. This reflects that the poly(A) on existing or newly synthesized mRNA is relatively protected from deadenylation in cells in which LARP4 is overexpressed (*Mattijssen et al., 2017*). In this model, the mRNA PATs are longer because they undergo less deadenylation. The PAT length of stable GFP mRNA bearing a short 3' UTR was dose-dependent on LARP4 levels (*Mattijssen et al., 2017*). PAT lengthening and stabilization were also observed for a β-globin reporter mRNA with an AU-rich element (ARE) that was previously known to be destabilized via recruitment of the CCR-NOT deadenylase complex (*Fabian et al., 2013*; *Mattijssen et al., 2017*). Endogenous mRNAs exhibited shorter PATs and decreased stability in LARP4 KO cells relative to LARP4 WT cells (*Mattijssen et al., 2017*). Full PAT net-lengthening and mRNA stabilization activity of LARP4 requires its PABP-interaction motifs albeit the PBM more so than the PAM2w (*Mattijssen et al., 2017*).

Although the LARP4 activities of PAT lengthening and mRNA stabilization appear to be coordinated (*Mattijssen et al., 2017*), a challenge has been to uncover their mechanistic link. Moreover, aspects of these activities may reflect more generally on relationships between PAT length and mRNA decay. For example, although LARP4 promotes lengthening of long PATs (*Mattijssen et al., 2017*) how this leads to mRNA stabilization may be revealing in the context of a model in which mRNAs are turned over only when their PATs become very short (*Chen and Shyu, 2017*; *Chen and Shyu, 2011*; *Eisen et al., 2020a*).

We developed a transcriptome-wide, SM-PAT-seq method that revealed a general effect of LARP4 on ~13,500 human and ~10,500 mouse mRNAs. Metagene analysis revealed that mRNA PAT length is increased by LARP4 in a dose-dependent manner and decreased by its genetic deletion, providing evidence that LARP4 is a general factor in mRNA PAT homeostasis. While LARP4 effects are clear on long PATs, its absence is most pronounced as PATs shorten to ≤75 nts. The data indicate that LARP4 shapes deadenylation profiles throughout the lifespan of mRNA, with most obvious impact when PAT length is in the size range known to affect mRNA turnover. We then used SM-PAT-seq for time course analysis of poly(A) decay which better resolved the differences between the LARP4 KO and WT profiles. The data provide new insight linking LARP4 to PAT protection from deadenylation and associated mRNA stabilization. An interpretation of our data in the context of recently published results led to a model of LARP4 action in which it protects against deadenylation and mRNA decay by contributing to the stabilization of PABP complexes on short PATs.

## Results

### Transcriptome wide, long-read, single-molecule poly(A)-tail, SM-PAT-seq

Pacific Bioscience Sequel technology uses single-molecule sequencing of long amplicons. *Figure 1A* outlines library preparation of circular templates. Each circularized DNA representing a single mRNA molecule with its full length PAT is replicated multiple times, both the sense strand and its complement (*Figure 1B*). Multiple 'subreads' generate one circular consensus sequence (CCS) read containing a unique gene-derived mRNA fragment followed by a nucleotide length-resolved PAT specific to the single mRNA molecule it represents (*Figure 1B*).

Small circles produce more copies per unit time than big circles, which generates higher CCS read accuracy. Thus, limiting amplicon length with random hexamers (*Figure 1A*) rather than directing 5'-extended cDNA produces high accuracy reads with full length PATs and ample gene-specific mapping information.

An objective comparison of SM-PAT-seq and RNA-seq was performed. The same RNA sample was analyzed for transcriptome-wide read counts by both methods and revealed an $R^2$ correlation = 0.58 with $p$ value of 2.2e-16 (*Figure 1—figure supplement 1A*). Assuming that RNA-seq is itself imperfect for quantitation and negatively impacted the $R^2$ value to some degree, the data are evidence that SM-PAT-Seq can indeed also serve to reflect relative mRNA abundances. Triplicate quantifications of GFP mRNA by SM-PAT-Seq and northern blotting produced an $R^2$ correlation = 0.96 with a p=0.000035 (next section). Finally, we note that >99% of SM-PAT-seq CCS reads reflect single molecule mRNAs as <1% CCS read identity was detected per library (*Figure 1—figure supplement 1B*).

### LARP4 expression leads to poly(A) tail net-lengthening of thousands of mRNAs

SM-PAT-Seq was performed on HEK293 cells expressing LARP4 at three levels, each in triplicate. Cells received empty plasmid or plasmid expressing LARP4 at ~3X or ~11X higher than endogenous

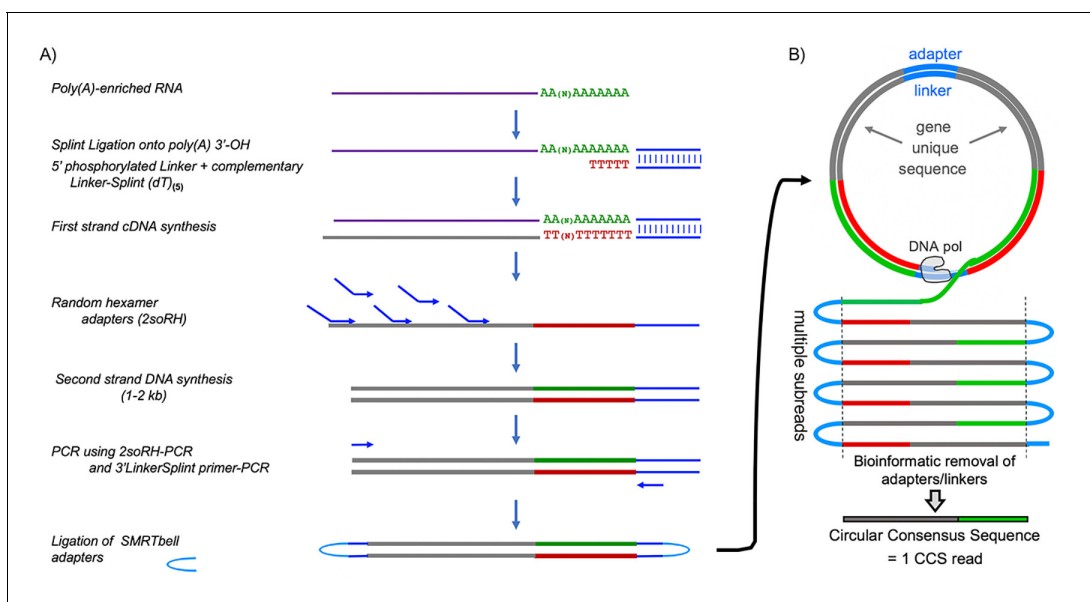

**Figure 1.** Flow diagram of SM-PAT-seq library preparation, sequencing and read output. (**A**) Overview of library construction for SM-PAT-seq, depicting production of a circular DNA from a single mRNA molecule with a specific poly(A) length. (**B**) Schematic production of one circular consensus sequence (CCS) read (bottom) from the mRNA-derived DNA template as performed by the Pacific Biosciences Sequel system (see text); DNA pol = DNA polymerase.

The online version of this article includes the following figure supplement(s) for figure 1:

**Figure supplement 1.** Analysis of SM-PAT-seq performance.

LARP4 levels (1X) (*Figure 2—figure supplement 1*), and each also received a uniform amount of GFP expression plasmid. *Figure 2A* shows effects on PAT lengths for ~13,500 nuclear-encoded mRNAs (includes poly(A)$^+$ transcripts annotated as long noncoding RNAs) (*Supplementary file 1*). Each SM-PAT-Seq CCS read represents an mRNA molecule with a specific PAT length. LARP4 affected a large number of mRNAs distributed across a wide range of PAT lengths (*Figure 2A*). LARP4 expression led to shifts in mRNA PAT length distributions in three length ranges (*Figure 2A*), 30–60 nt, 60–90 nt, and 90- > 250 nt. At 11X, the fraction most increased was mRNAs with long PATs, 150–250 nt. The fraction shifted with 3X LARP4 was also to longer PATs, but less than with 11X. As *Figure 2A* shows fractions of total reads, an increase in one size range is associated with decrease in another. The intersection of the 1X and 11X tracings was at 110–90 nt.

An interpretation of *Figure 2A* is that an increase in LARP4 levels leads more mRNAs to persist with long PATs at the expense of accumulation of mRNAs with short PATs. Increasing LARP4 alters the mRNA population such that more have longer PATs and less have shorter PATs. Given current models, increasing LARP4 may slow the conversion of long PAT-mRNAs to short PAT-mRNAs.

Median mRNA PAT lengths of all genes observed with a minimum of 10 CCS reads are shown as violin plots in *Figure 2B*. Thousands of mRNA PATs are shifted to longer lengths by increasing LARP4 levels. The median PAT length 50% cumulative read fraction was shifted from ~75 nt with 1X LARP4 to ~82 and~90 nt at 3X and 11X LARP4, respectively (*Figure 2C*).

The average PAT lengths for functionally related gene subsets as determined by gene ontology (GO) were almost all correlated with LARP4 expression levels, with the one clear exception being the mitochondrial DNA-encoded mRNAs, as expected (*Figure 2D*). However, an unexpected exception was the set of nuclear-encoded mitochondrion mRNAs. The PATs on the mRNAs that comprise this set are significantly shorter than on most other mRNAs in cells with 1X LARP4, and their lengths increased less in response to higher LARP4 levels relative to the PAT length increases in the other mRNA GO sets. This reflects a highly significant difference including by comparison to the RNA binding GO set as both are comprised of ~2000 mRNAs and represented by 35,000–50,000 CCS read counts in each LARP4 condition.

Prior analysis provided indirect evidence of PAT lengthening of GFP mRNA by LARP4 (*Mattijssen et al., 2017*). Here, all cells were transfected with equal amounts of GFP plasmid and triplicate samples were analyzed by northern blot and SM-PAT-Seq. This showed that GFP mRNA accumulation and PAT lengthening was dose-dependent on LARP4 (*Figure 2E,F*). As LARP4 levels increased, GFP mRNA median PAT lengths increased and were accompanied by increases in GFP CCS reads (*Figure 2F*), reflective of the trends of GFP mRNA levels and mobility shifts seen by northern blot (*Figure 2E*). Comparison of GFP CCS quantitative reads and GFP mRNA quantitation by northern revealed a R$^2$ value of 0.96 with p=0.000035 (*Figure 2G*). The general likeness of band intensity distribution in the northern blot 1X, 3X and 11X lanes, and shapes of the corresponding violin plots suggests that LARP4-mediated protection from deadenylation of long PATs specifically, confers the stable GFP mRNA with greater accumulation/stability (Discussion).

## Absence of LARP4 leads to poly(A) tail shortening of thousands of mRNAs

The left panel of *Figure 3A* shows SM-PAT-Seq analysis of ~10,500 nuclear-encoded mRNA CCS reads from triplicate MEF cell lines (*Supplementary file 2*). The KO MEFs have more mRNA CCS reads with short PATs (30–60 nt) than with longer PATs (75–200 nt) as compared to WT (*Figure 3A*, left panel). The cumulative fraction plots (right) show an overall length shift to shorter PATs in the KO MEFs. Violin plots (*Figure 3B*) show that the median PAT lengths of KO MEF mRNAs are shorter than in WT MEFs, for thousands of mRNAs. In contrast to nuclear DNA gene-encoded mRNAs, LARP4 expression had no apparent effect on mitochondrial DNA-encoded mRNA PATs (*Figure 3—figure supplement 1*).

## Alteration of a phasing pattern of mRNA poly(A) tails in the absence of LARP4

Occurrence of sequential peaks in the poly(A) length profiles of efficiently translated mRNAs such as those that encode ribosomal proteins is referred to as phasing (*Lima et al., 2017*). The phased PAT length peaks derive from fluctuations in the rates of deadenylation around sites protected by serial

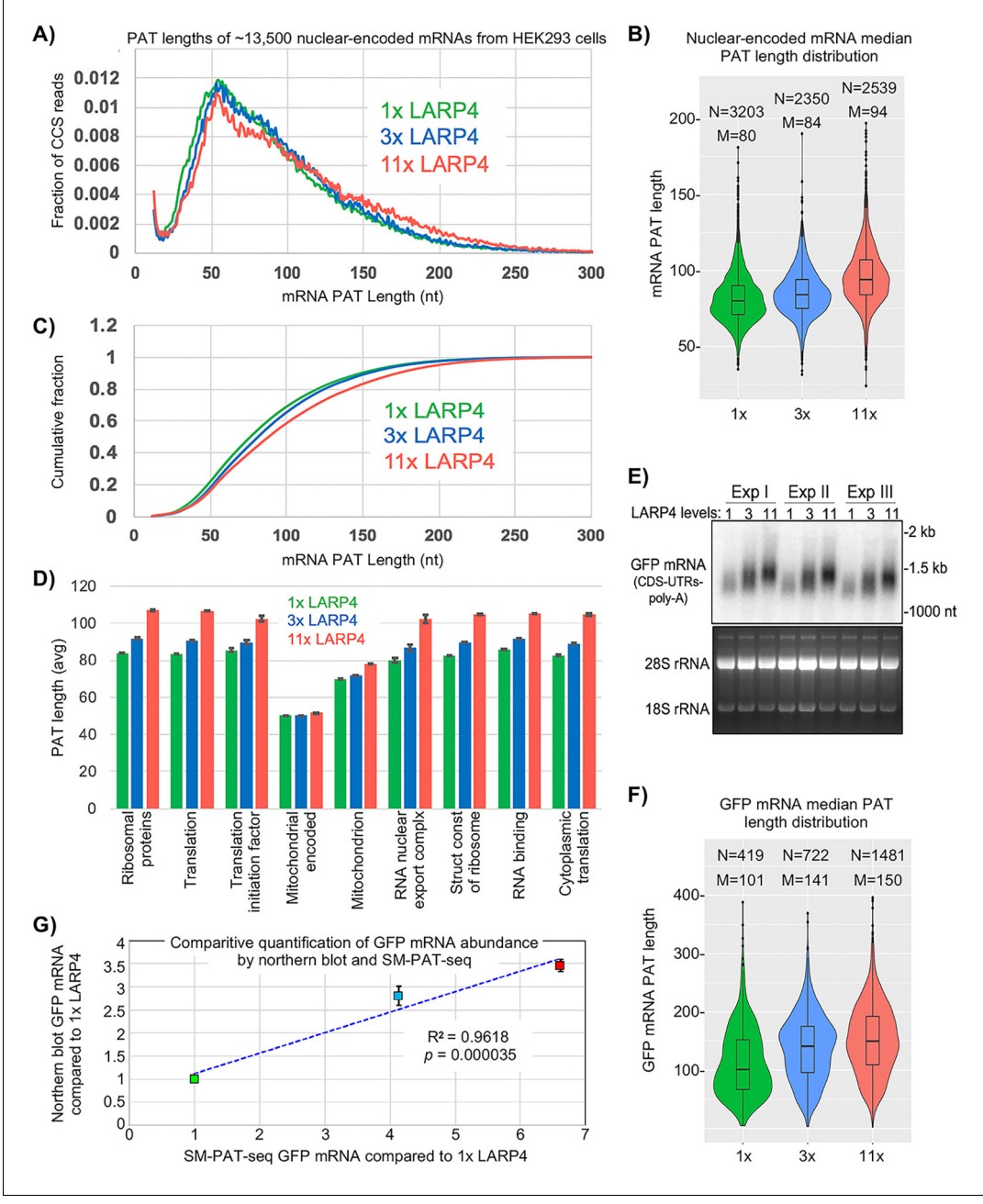

**Figure 2.** SM-PAT-seq shows LARP4 overexpression increases global PAT length in HEK293 cells. (**A**) Fraction of total reads vs. PAT length, at three levels of LARP4 expression, comprising ~13,500 annotated gene transcripts, with ~200,000 CCS reads per condition. (**B**) Violin plots of median PAT lengths determined for genes with ≥10 CCS reads. N: number of genes, M: median length. The *p* values calculated using a 2-tailed Welch's T-test with unequal variance are as follows; 1x vs 3x: 4.81E-17, 3x vs 11x: 3.94E-102, and 1x vs 11x: 3.10E-186. (**C**) The data in C plotted as cumulative distribution, read fraction sum at or below PAT length x. Analyses in A-C excluded mitochondrial-encoded transcripts. (**D**) Average PAT lengths for various functional group mRNAs as determined by gene ontology (GO) analysis, error bars represent the standard error. (**E**) Northern blot analysis of the triplicate RNA samples used for SM-PAT-seq probed for GFP mRNA. Bottom is gel prior to transfer. (**F**) GFP mRNA median PAT-length by SM-PAT-Seq. N: number of GFP CCS reads, M: median length. The *p* values calculated for the GFP mRNA median PAT lengths using a 2-tailed Welch's T-test with unequal variance are as follows; 1x vs 3x: 2.72E-10, 3x vs 11x: 3.60E-8, and 1x vs 11x: 4.01E-27. (**G**) Comparison of the triplicate GFP CCS reads and the triplicate GFP mRNA quantitation by northern blot for1X, 3X and 11X LARP4 levels revealed a R$^2$ = 0.96 with a *p* value of = 0.

*Figure 2 continued on next page*

*Figure 2 continued*

000035 calculated using Fisher transformation of the correlation coefficient and the sample size; the error bars reflect standard deviation of the Northern data.

The online version of this article includes the following figure supplement(s) for figure 2:

**Figure supplement 1.** Quantification of LARP4 expression levels in HEK293 cells analyzed by SM-PAT-seq in biological triplicate experiments I, II and III.

PABPs bound at those positions (*Lima et al., 2017*; *Nicholson and Pasquinelli, 2019*) (see *Yi et al., 2018*).

It was previously shown that three ribosomal protein (rp) mRNAs have shorter PATs and shorter half-lives in LARP4 KO MEF cell lines as compared to WT MEF cell lines (*Mattijssen et al., 2017*). *Figure 3C* shows SM-PAT-seq data from the rp-mRNAs as a subset of the MEF total mRNAs. The left panel shows that both profiles revealed a bimodal distribution in the ~35–70 nt size range, although the relative heights of the two peaks clearly differed in WT and KO. The longer PAT peak was higher in the KO cells and associated with a steeper transition incline from long to short PATs in the KO (*Figure 3C*). Interestingly, similar to *Figures 2A* and *3A*, the transitions from long to short

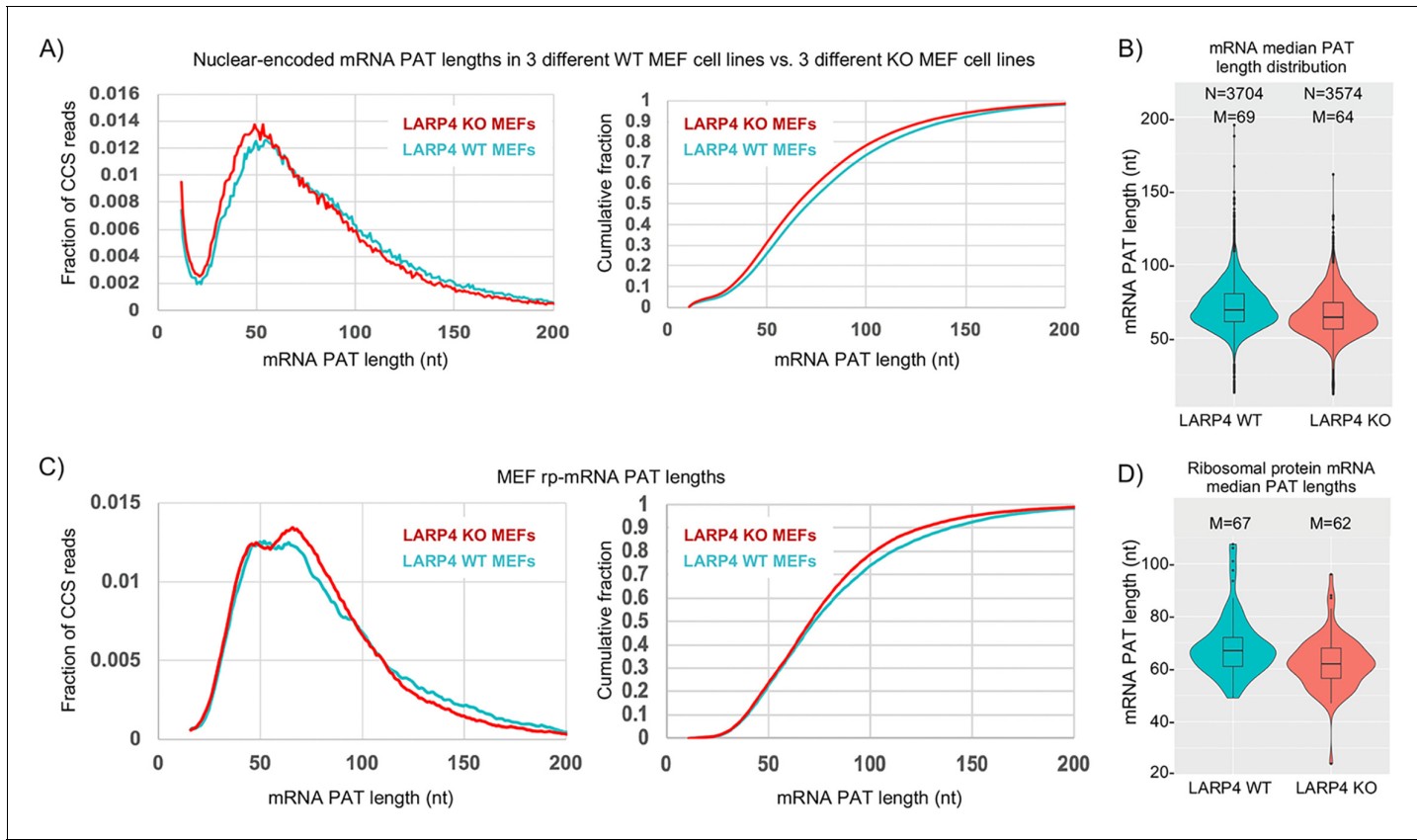

**Figure 3.** LARP4 deletion decreases mRNA PAT length in MEFs. (**A**) Left: Fraction of total CCS reads vs. PAT length for triplicate WT vs. LARP4 KO MEFs for ~10,500 nuclear-encoded mRNAs. Right: The same data plotted as cumulative distribution. The *p* value for WT vs KO calculated using a 2-tailed Welch's T-test with unequal variance was 4.53E-291. (**B**) Violin plots of median PAT length distribution per gene as described for *Figure 2B*. The *p* value for WT vs KO median PAT lengths calculated using a 2-tailed Welch's T-test with unequal variance was 9.34E-55. (**C**) Left: Fraction of total CCS reads vs. PAT length for a subset of 77 ribosomal protein-mRNAs data from A. The *p* value for WT vs KO rpmRNAs calculated using a 2-tailed Welch's T-test with unequal variance was 1.81E-61. Right: The same data plotted as cumulative distribution. (**D**) Violin plots of median PAT length distribution per rp-mRNA as described for *Figure 2B*, on the rp-mRNAs in panel C with ≥10 CCS reads. The *p* value for WT vs KO median PAT lengths for rp-mRNAs was calculated using a 2-tailed Welch's T-test with unequal variance was 0.000431.

The online version of this article includes the following figure supplement(s) for figure 3:

**Figure supplement 1.** SM-PAT-seq profiles of mitochondrial DNA encoded mRNAs.

PATs also differed for KO and WT and overlapped in the ~110–90 nt size range f the rp-mRNAs (Discussion). We conclude that genetic deletion of LARP4 alters the PAT length phasing pattern of rp-mRNAs in MEFs in the ~70–35 nt size range, and this is associated with differences in profile shapes of longer PATs.

## LARP4 promotes accumulation of interferon-induced mRNAs

An ARE in LARP4 mRNA mediates decreased LARP4 levels in response to the cytokine tumor necrosis factor-α (TNFα) (*Mattijssen and Maraia, 2016*). The type I interferon (IFN)-stimulated genes (ISGs) are induced by a variety of molecules including some interleukins, TNFα, and cytokines that activate various extracellular receptors (*Hergovits et al., 2017*; *Wang et al., 2016*). ISGs are also induced by activation of *intra*cellular receptors in response to viral infection and other pathogens; different types of nucleic acids activate distinct pathways to ISG induction (*Atianand and Fitzgerald, 2013*; *Schneider et al., 2014*). Triplicate RNA-seq revealed that a set of 128 ISG mRNAs were induced to ≥1.5 fold higher levels with a Padj-value <0.05 in HEK293 cells expressing 3X LARP4 relative to 1X after 1 kb AT-rich DNA transfection (*Supplementary file 3*). We cotransfected 293 cells with LARP4 at 3X levels together with the dsRNA poly(I:C), (dA:dT)(80), or 1 kb AT-rich DNA, and monitored expression of the ISG mRNAs, IFIT1 and ISG15 (*Figure 4A*, lanes 1–20). Triplicate data for the dsRNA poly(I:C) ISG induction are shown in *Figure 4A*, lanes 1–12 and quantified in *Figure 4B*. This verified that higher levels of ISG mRNAs accumulated with elevated LARP4 (+)

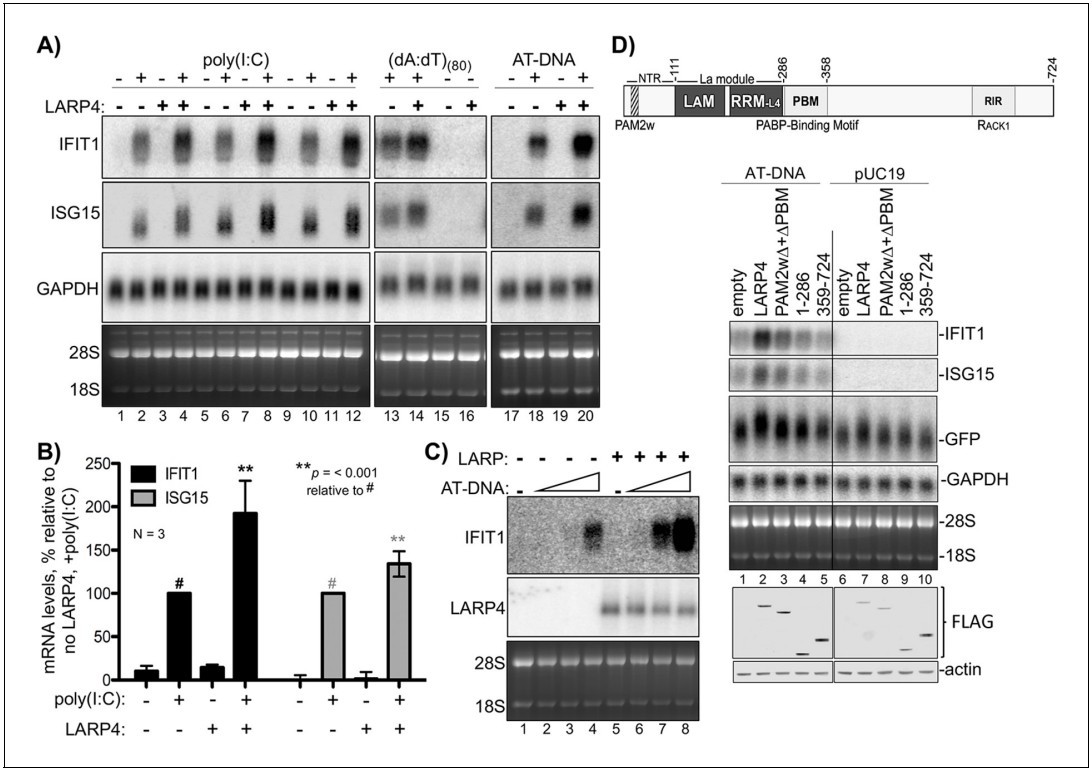

**Figure 4.** LARP4 promotes accumulation of interferon-induced innate immune mRNAs. (**A**) Northern blot analysis of two ISG mRNAs, IFIT1 and ISG15 after transfection-mediated induction by IFN-stimulating nucleic acids, and + / - co transfection with LARP4, as indicated above the lanes. Bottom panels are stained gels prior to transfer. (**B**) Quantitation of biological triplicate northern blot data, normalized by GAPDH, with the poly(I:C) +, LARP4 - sample (annotated with # above the bar) set to 100%; error bars represent 95% confidence interval. *P* values were calculated using a 2-tailed Welch's T-test with unequal variance. (**C**) Northern blot of IFIT1 and LARP4 from total RNA 48 hr after transfection of 0, 2, 4 or 6 μg of 1 kb AT-rich DNA in 6-well format (the total DNA amount for this component of the transfection was maintained at 6 μg with carrier pUC19), plus 2.5 μg pFLAG-LARP4 (+) or empty pFLAG vector (-) as indicated. (**D**) Top: schematic of LARP4 showing its two PABP-interaction motifs, PAM2w and PBM, the La-module comprised of LaM (La motif) and RRM-L4, and the RACK-1 interaction region (RIR). The N-terminal region (NTR) which is responsible for poly(A)-binding (*Cruz-Gallardo et al., 2019*), is also indicated. Middle: Northern blot after co-transfection of 1 kb AT-rich DNA or pUC19 and various LARP4 constructs indicated above the lanes. Bottom: Western blot analysis of protein from the same cells.

expression (*Figure 4B*). Slower mobility was observed in the ISG15 mRNA from the cells transfected with LARP4 (+), indicative of longer PATs (*Figure 4A*, lanes 2 and 4, 6 and 8, 10 and 12).

Transfection of LARP4 also led to higher mRNA levels after treatment with ISG-inducing DNAs, $(dA:dT)_{(80)}$, or 1 kb AT-rich DNA (*Figure 4A* lanes 13–16, 17–20). As noted, LARP4 levels can change in response to external stimuli, e.g., by TNFα (*Mattijssen and Maraia, 2016*). We tested if increasing LARP4 levels (3X) could sensitize HEK293 cells to ISG-inducing DNA (*Figure 4C*). For this, cells were transfected with empty plasmid (lanes 1–4) or equal amounts of LARP4 expression plasmid (lanes 5–6) and varying amounts of 1 kb AT-rich DNA, the latter as indicated by triangles above the lanes (*Figure 4C*). The cells expressing 3X higher levels of LARP4 produced more ISG mRNA at a lower dose of inducing DNA. Thus LARP4 expression can contribute to the levels to which ISG mRNAs accumulate after their induction (*Figure 4C*).

Ability to see increased ISG mRNA accumulation in response to LARP4 provides an assay for induced endogenous transcripts. Examination of different LARP4 constructs revealed that increased accumulation of IFIT1 and ISG15 mRNAs is dependent on the PABP-interaction motifs PAM2w and PBM, of LARP4, also required for GFP mRNA mobility shift indicative of PAT protection (*Figure 4D*; *Mattijssen et al., 2017*).

## LARP4 slows PAT shortening during mRNA lifespan

To examine PAT length dynamics, ISG mRNAs were induced in KO and WT MEFs by treatment with IFNα, followed by actinomycin D (ActD) to inhibit transcription, and RNAs were subjected to SM-PAT-seq at times thereafter to monitor deadenylation. Separate analyses were done on ISG mRNAs and on total mRNAs.

As part of the same duplicate experiments, northern blots of total RNA revealed a ~ 4 hr half-life for ISG15 mRNA in WT cells, consistent with prior data (*Li et al., 2000*), but 2.8 hr in KO cells, reflecting lower stability (*Figure 5A,B*). The ISG15 mRNA bands exhibited progressively faster mobility with time after ActD reflective of deadenylation (*Figure 5A*).

PAT shortening after ActD treatment was apparent for the total mRNA set with clear differences between LARP4 WT and KO (*Figure 5—figure supplement 1A*). However, there were relatively large spans for the 95% confidence intervals represented by the whisker plots (*Figure 5—figure supplement 1A*).

To better resolve these data, they were plotted as the fraction of CCS reads vs. PAT length at each time point (*Figure 5C*). A notable feature is the transition from PAT length of ~100 nt to the peak at ~75 nt for the KO and WT tracings at time zero (*Figure 5C*, t = 0). Although both time zero profiles were very similar with the majority of mRNA PATs at 25–75 nt, this fraction was slightly higher in amplitude in KO than WT (panel t = 0).

A more striking feature was observed at 30 min as a significant fraction of mRNAs appeared with shorter PATs, with a peak at 35–40 nt in the KO profile, while no such peak fraction was distinguished in the WT profile (*Figure 5C*, panel t = 30). By 60 min the WT profile showed a fraction of mRNA PATs of ~35–40 nt similar to what appeared earlier in the KO profile.

A substantial amount of the total mRNA mass shifted to shorter PATs during the time course. The t = 0 peak at ~75 nt shrank with time as the 35–40 nt peak enlarged, as expected of a precursor-product relationship (compare t = 60, 120, 240, 480). This would appear to reflect redistribution of phased PABP protection of mRNA PATs during deadenylation that accompanies decay.

The ~75 nt peak seen in *Figure 5C* is consistent with mRNAs that would harbor three PABP molecules (*Webster et al., 2018*) (Discussion). By the t = 720 min time, a single ~38 nt peak is in the KO profile while the WT profile retains remnants of the precursor ~75 nt peak. Thus, conversion of the distinct mass of the ~75 nt PAT peak to ~38 nt PATs was initiated faster and completed more readily in LARP4 KO than in WT cells.

The time course PAT decay data in *Figure 5C* provide evidence that the presence of LARP4 slows deadenylation and this becomes most apparent at the point beyond which mRNAs accumulate with poly(A) lengths of ~75 nt. As discussed later, recent data suggest that the ~75 nt peak represents mRNAs to which three PABPs are bound and undergo deadenylation during the time course.

We identified a set of 194 ISG mRNAs by inducing them in MEFs (*Supplementary file 4*) and examined their PAT shortening at times after ActD. Each box plot in (*Figure 5—figure supplement 1B*) comprises >5000 CCS reads representing multiple ISG mRNA transcripts that span a range of PAT lengths that significantly differ in LARP4 WT and KO. To resolve these, the reads were sorted

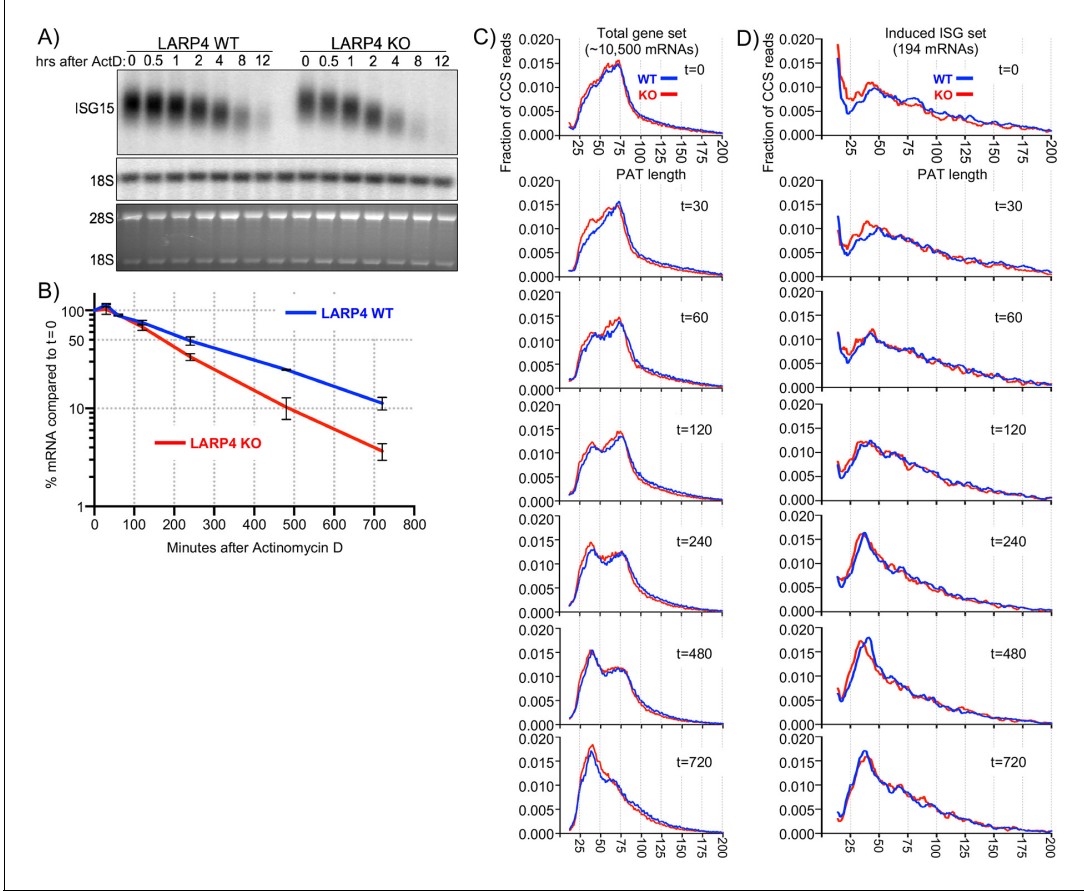

**Figure 5.** SM-PAT-seq decay analysis of total and ISG mRNAs. Duplicate sets of LARP4 KO and WT MEFs were treated with IFNα, followed by transcription inhibition with actinomycin D, RNA isolation at multiple times thereafter, and analysis as follows. (**A**) Northern blot probed for ISG15 mRNA and 18S rRNA (top panel); stained gel prior to transfer (bottom panel). (**B**) Quantitation of biological duplicate northern data. ISG15 signal was normalized to 18S rRNA; t = 0 was set to 100%, error bars represent the spread. (**C**) Fraction of SM-PAT-seq CCS reads vs. PAT length for each time point after ActD addition for the gene set of ~10,500 mRNAs as in *Figure 3A*. (**D**) Fraction of SM-PAT-seq CCS reads vs. PAT length for each time point for the ISG set of 194 mRNAs. For C and D, the mitochondrial-encoded reads were filtered out.

The online version of this article includes the following figure supplement(s) for figure 5:

**Figure supplement 1.** Time course analysis of deadenylation by SM-PAT-seq.

by PAT size into ten bins; those with the longest in the top 10% length bin and so on, with each time point also displayed in each bin (*Figure 5—figure supplement 1C*). By this approach, the greatest variability in PAT length sorted with the longest PATs, consistent with recent observations (*Eisen et al., 2020b*). PAT shortening was observed in all of the incremental size ranges (bins) in WT and KO except the last/shortest.

The fraction of total CCS reads for ISG mRNAs vs. PAT length at times after ActD was plotted (*Figure 5D*). These profiles differ from those of the total mRNA profiles in their overall shapes at t = 0 (*Figure 5C*). This is most notable in the 30–90 nt range, which is of lower amplitude in the ISG profile, although it substantially increases with time as expected for a set of induced ISGs vs. total RNAs. Reciprocally, the fraction of longer PAT reads is larger in the ISG than in the total gene set at t = 0, reflective of induced mRNAs.

At t = 0 the ISG peak at 35–40 nt for the KO tracing is slightly to the left and higher on the Y-axis than the WT peak, which is below the 0.010 fraction level, suggesting relative accelerated PAT shortening in the KO cells during the IFN treatment period (*Figure 5D*). At t = 30 min the KO tracing has more clearly accumulated reads in the ~38 nt peak. At t = 60 and t = 120 both the KO and WT tracings accumulate more mass above 0.010 in the 24–50 nt PAT size. By t = 240 min the ~38 nt peaks

have sharply risen above the 0.015 fraction mark, with the KO tracing leading toward shorter PATs (*Figure 5D*).

These data revealed deadenylation kinetics of 194 ISG mRNAs and the larger transcriptome of ~10,500 mRNAs. They showed that shortening of mRNA PATs to ones that accumulate with ~38 nt PATs occurred more readily in LARP4 KO than in WT cells. Thus, by following mRNA PAT lengths over time, deadenylation was observed to occur with less impediment in the absence of LARP4 than in its presence.

## Discussion

LARP4 is a mRNA stabilizing protein that can bind to poly(A) via its N-terminal region (NTR) which involves the PAM2w, and also binds PABP via another motif (*Cruz-Gallardo et al., 2019*; *Yang et al., 2011*). We developed a single molecule mRNA sequencing method for measuring and quantifying PAT lengths transcriptome-wide to study effects of LARP4 on poly(A) metabolism. A major conclusion is that LARP4 is a general factor that slows deadenylation transcriptome-wide, impacting a large fraction of mRNAs. For HEK293 cells and MEFs, LARP4 protects PATs from deadenylation over a wide spectrum of lengths, long and short. By examining PAT length after inhibition of transcription we conclude that mRNAs are deadenylated faster in KO than in WT cells. The data indicate that PATs of ≤75 nt are more susceptible to deadenylation in the absence of LARP4 than in its presence. The functional relevance of this is clear as LARP4 is an mRNA stabilization factor (*Mattijssen et al., 2017*). Data in *Figures 2A* and *3A–C* provide evidence that LARP4 slows conversion of long PAT- to short PAT-mRNAs. The PATs of the stable GFP mRNA expressed in transiently transfected HEK293 were longer than the cellular mRNAs (*Figure 2B,F*) likely reflecting that they are newly transcribed whereas the latter comprise a steady state population of mixed species (*Eisen et al., 2020b*). Nonetheless LARP4 expression led to dose-dependent increases in their PAT lengths. More importantly, SM-PAT-seq and northern blot data for GFP mRNA showed that increased PAT length was associated with their greater accumulation. It is notable that increasing LARP4 activity protected long PATs on GFP mRNA as reflected by northern and violin plots (*Figure 2E,F*). Thus, it would appear that LARP4 can increase the stability of this stable GFP mRNA by protecting against deadenylation during its lifespan when PATs are long.

Multiple mechanisms can control LARP4 levels over a wide range, including via the pro-inflammatory cytokine TNFα (*Mattijssen et al., 2017*; *Mattijssen and Maraia, 2016*). We showed that induced ISG mRNAs accumulated to higher levels when LARP4 was modestly increased. Reciprocally, ISG15 mRNA decayed faster in LARP4 KO than in WT cells accompanied by evidence of PAT shortening.

LARP4 can stabilize both stable mRNA and unstable mRNA (*Mattijssen et al., 2017*). Thus, it is possible that a change in LARP4 levels at a single time point can differentially affect turnover of different sets of mRNAs at different times thereafter, depending on characteristics of their lifespan poly(A) profiles.

### Altered phasing of rp-mRNA PATs; evidence that LARP4 impedes deadenylation by CNOT

Biochemical and structural studies of yeast factors indicate that Pan2-Pan3 can efficiently trim long PATs with serially bound PABP in part because of a unique architecture of PABP-PABP interactions on poly(A), whereas Pan2-Pan3 is inefficient on short poly(A)-PABP mRNPs (*Schäfer et al., 2019*). Biochemical studies using human factors showed that although PABP recruits the CNOT complex to promote deadenylation of shorter PATs, the CCR4 and CAF1 deadenylase subunits respond very differently (*Yi et al., 2018*), consistent with studies using the yeast homologs (*Schäfer et al., 2019*; *Webster et al., 2018*). The Caf1 subunit of CNOT can deadenylate unprotected poly(A) but the Ccr4 subunit is required to dislodge or peel off PABP (*Webster et al., 2018*; *Yi et al., 2018*). These and other findings help explain some aspects of irregular deadenylation noted during mRNA decay (*Chen et al., 2017*; *Webster et al., 2018*; *Yamashita et al., 2005*; *Yi et al., 2018*).

PAT length phasing is related to nonuniform deadenylation around PABP sites and was shown to be more sharply prominent in human cell lines whose CCR4 levels were decreased (*Yi et al., 2018*). While the ascending and descending slopes of a phased PAT peak would appear to reflect different deadenylation rates before and after PABP dissociation from a binding site, the larger overall shape

of a PAT profile can reflect additional irregularity (*Yi et al., 2018*). The biochemistry indicates that deadenylation progressively accelerates as bound PABP decreases to two then to one molecule and thereafter (*Webster et al., 2018*; *Yi et al., 2018*). The decrease in stability of PABP complexes would appear to result in part from architectural changes as PATs shorten and from loss of PABP-PABP intermolecular contacts (*Schäfer et al., 2019*; *Webster et al., 2018*).

Although offsets in PAT peaks on different individual genes can undermine phasing in metagene profiling (*Yi et al., 2018*), SM-PAT-seq revealed clear differences in the phased peaks of the collective rp-mRNAs from LARP4 KO and WT cells (*Figure 3C*). Here we propose a model view of our PAT phasing results in *Figure 6*. First, we note that the rp-mRNA profiles differ most dramatically in that the KO peak that spans ~60–75 nt contains a greater fraction of reads than the WT (*Figure 6A*). Second, this higher peak is associated with a transition from long to short PATs that is of steeper incline in KO than WT, annotated by the upward red arrow. Third, the WT and KO transitions from long to short PATs intersect at ~110 nt (inverted triangle), a length known to be associated with transition from deadenylation of long PATs by PAN2/3 to shorter PATs by CNOT (*Yamashita et al., 2005*; *Yi et al., 2018*).

We suspect that the KO 60–75 nt peak represents PATs bound by three PABPs that include peeling off/dissociation intermediates (*Webster et al., 2018*; *Figure 6C*), that are being shortened to accumulate in the peak of ~40–50 nt PATs bound by two PABPs. By extrapolation, PATs bound by two PABP-*dimers* would appear in the approximate 100–110 nt range in which the KO and WT profiles overlap (*Figure 6A*). The steeper incline of the KO transition profile might suggest more efficient formation of the 60–75 nt PATs, consistent with LARP4 impeding deadenylation by CNOT. Also consistent with this is the more irregular shape of the ascending WT profile. Likewise, different shapes of the peaks suggest they may represent different intermediates in the presence and absence of LARP4. In any case, it would appear that formation of $PABP_{(3)}$-PAT complexes by CNOT may occur more readily in the absence of LARP4 (see *Figure 6A* legend). It is also possible that differences may reflect the presence/absence of proteins other than PABP.

Kinetic profiling affords additional considerations (*Figure 6B*). At time zero the major PAT peak was 75 nt in KO and WT. Further deadenylation occurred sooner thereafter in KO cells as shorter peaks appeared at 30 min (arrows) that had not yet appeared in WT. The overall profile kinetics suggest that the proposed ~75 nt $PABP_{(3)}$ complex (*Figure 6C*) is less stable in LARP4 KO than in WT cells. An intermediate (dashed arrow) appeared between the 75 and 38 nt (solid arrow) peaks in the KO profile at 30 min that was not distinguished in WT (also see *Figure 5C*). Evidence of this is also in the 60 min KO profile more broadly (bracket) and may reflect a deadenylation intermediate in LARP4 KO cells whose formation is otherwise blocked/inhibited or masked by LARP4. *Figure 6C* depicts schematized $PABP_{(3)}$-$PABP_{(2)}$ PAT intermediates.

These analyses have advanced insight into the role of LARP4 action in PAT lengthening and associated mRNA stabilization. Our findings in the PAT size range in which deadenylation is known to promote mRNA decay in mammalian cells provide a mechanistic link of the activities of PAT 3' end protection and mRNA stabilization of LARP4. It was proposed that LARP4 could protect PATs from deadenylation by RNA 3' end binding (*Mattijssen et al., 2017*). The data here leave open the possibility that LARP4 may directly protect RNA 3' ends, and also support a model in which it stabilizes PABP on short PATs to shield them from deadenylases. In this model, decreased PABP-PAT stability would be compensated by interactions with LARP4 (*Cruz-Gallardo et al., 2019*; *Mattijssen et al., 2017*; *Yang et al., 2011*). Such interactions are likely intricate as LARP4 can employ two motifs for PABP binding including by its unique PAM2w which can also be used to alternatively bind poly(A) (*Cruz-Gallardo et al., 2019*). As noted, PAM2 peptides that associate with PABP via its MLLE domain are also found on PAN3 as well as Tob1 and Tob2 which interact with the CAF1 subunit of CNOT. Thus, in the milieu of the cellular deadenylation machinery, the PAM2w would appear to contribute to an intricate network of factors in competition for poly(A) and PABP.

CNOT is an intricate multimeric complex with much regulatory potential for general and mRNA-specific deadenylation (*Raisch et al., 2019*), refs therein). Effects on long PATs and stable mRNA demonstrated here suggest the possibility that LARP4 can influence mRNA early in its lifespan with penetrant consequence later. Thus this work on LARP4 raises issues regarding the long and short of PAT metabolism.

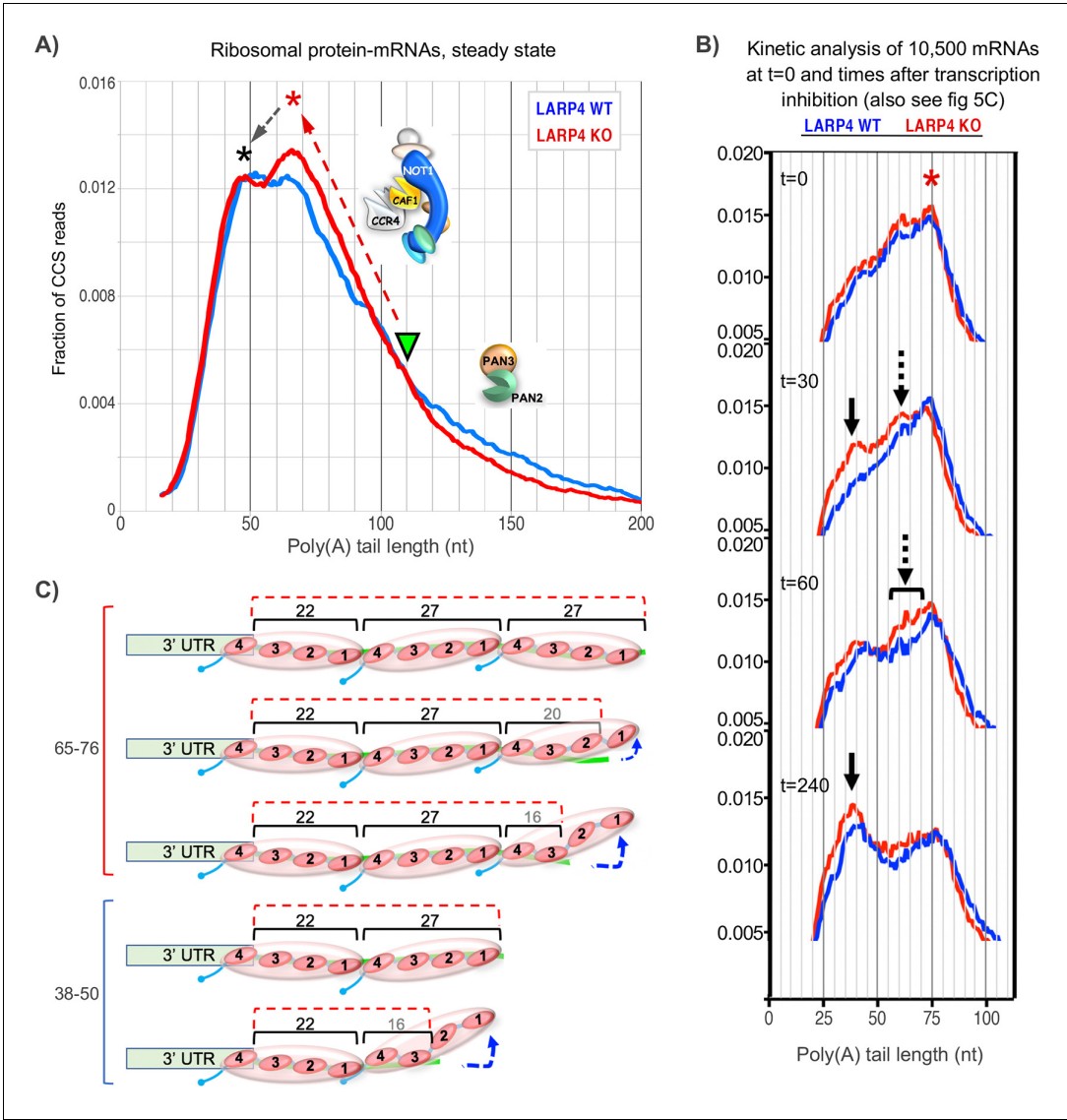

**Figure 6.** Proposed working models for differential deadenylation profiles in LARP4 KO and WT MEFs (derived from *Figure 3A* and select panels of 5C). (**A**) LARP4 effects on phasing of a set of ~80 ribosomal protein mRNAs at steady state was revealed by plotting the fraction of CCS reads vs. PAT lengths. A green triangle points to where the KO and WT transitions from long to short PATs intersect, at ~110 nt. The upward red dashed arrow denotes a steeper incline in KO as compared to WT of long to short PAT length transition. The red asterisk denotes the ~60–75 nt PATs whose peak is comprised of a substantially higher fraction of reads in KO as compared to WT cells. The black dashed arrow denotes conversion of longer to shorter PAT complexes in the peaks denoted by the corresponding asterisks. In this model, the peak denoted by the red asterisk would correspond to PATs bound by three PABPs that include peeling off/dissociation intermediates schematized in the upper part of panel C, and the peak denoted the black asterisk would correspond to PATs bound by two PABPs. As the two profiles transition from long to shorter PATs they begin to overlap at PAT lengths of ~110 nt which would correspond to mRNP complexes containing four PABPs or two PABP-dimers. These profiles suggest that as PATs shorten from this transition point and the distal PABP would begin to dissociate, CNOT may be less impeded in the absence of LARP4 (see text). Thus LARP4 may exert a stabilizing effect at this point. (**B**) Transcriptome-wide dynamic PAT phasing observed in SM-PAT-seq profiles in *Figure 5C* illustrating a deadenylation time course after transcription inhibition by actinomycin D. Red asterisk denotes the 75 nt PAT peak at t = 0; panels depicting t = 30, 60 and 240 min are also shown as labeled. Solid black arrow denotes a ~ 38 nt PAT peak, dashed arrows and bracket denote proposed intermediate (see text and *Figure 5C*). (**C**) Schematized depiction of potential PAT-PABP$_{(3)}$ and PAT-PABP$_{(2)}$ complexes and intermediate binding forms thereof that might comprise the mRNA-PAT profile peaks in A

*Figure 6 continued on next page*

*Figure 6 continued*

and B, in various states of peeling off modes of PABP dissociation, modeled after *Webster et al., 2018* and discussed in the text.

## Materials and methods

### SM-PAT-Seq

PolyA-selected RNA was generated from 4 µg of total RNA using poly-T oligo attached magnetic beads from illumina TruSeq Stranded mRNA kit. An adapter was ligated to the polyadenylated 3' end of the RNAs using T4 RNA Ligase 2 (NEB) with a double-stranded splint linker containing an overhanging stretch of 5 Ts. Following clean up, reverse transcription was performed using Superscript II (ThermoFisher). The RNA strand was digested using RNase H (NEB). The sample was cleaned up using an RNA Clean up kit (Zymo). Second strand synthesis was primed using random hexamers attached to an adapter sequence for PCR and extended using Klenow Fragment 3'−5' exo- (NEB). The ds-cDNA was purified using 0.6X Ampure PB beads (Pacific Biosciences).

The polyA tails with adjoining 3'UTRs were amplified using primers directed against distal ends of the splint linker (polyA end) and the adapter/random hexamer oligo. The PCR reaction employed KAPA HiFi Hot Start Ready Mix (Roche) for 25 cycles. The products were again purified using 0.6X Ampure PB beads (Pacific Biosciences). Primers and adapters are in *Supplementary file 5*. Resulting DNA amplicons were quantified using Qubit two and fragment size determined (1700 bp avg) using a high sensitivity DNA chip on a Bioanalyzer 2100 (Agilent), and prepared for sequencing using a SMRTbell Barcoded Adapter Complete Prep kit (Pacific Biosciences). Libraries were loaded onto a SMRT Cell V2 using diffusion loading and sequenced on a PacBio Sequel using 10 hr run-time.

### PAT dataset handling

Subread data consisting of a minimum of 3 passes from the PacBio Sequel was converted into CCS using the SMRT Analysis command line toolset and aligned using BLAT against the GENCODE database of full length mRNA sequences (vM16 mouse, v27 human) for assignment to originating mRNA. In addition, each sequence was evaluated for occurrence of poly(A) abutting adapter sequence in either strand direction. PAT length was assigned using the observed poly(A) while allowing for any number of single base disruptions provided that non-A was bracketed by a minimum of 2 A bases on either side as part of the longer continuous poly(A) tract. Reads were accepted as properly assigned and resolved provided they had a PAT longer than 10 bp and a successful alignment to a GENCODE v23 transcript with a BLAT score exceeding a minimum threshold of 50. For the rp-mRNA PAT profiles in Fig 3C the fractional data were smoothed by a sliding +/- 5 nt length averaging prior to plotting.

Gene assigned PAT lengths based on GO annotation (*The Gene Ontology Consortium, 2019*; avg. length and standard error were calculated and plotted for each subset in each LARP4 expression condition. The 194 MEF ISG mRNAs were those that exhibited ≥2 fold increase in levels in uninduced vs. t = 0 after IFNα treatment, with adjusted p-value of less than 0.1 (*Supplementary file 4*). Binning of PAT lengths was achieved using the top_frac function on length sorted data from the R package dplyr (*Wickham et al., 2019*) to produce 10 bins, using all of the non-mitochondrial assigned reads.

### Quantitative comparison of RNA-seq and SM-PAT-seq

Analysis of libraries was from the same MEF RNA sample but prepared either for RNA-seq or SM-PAT-seq. Following RNA-seq, transcripts were aligned with Salmon (*Patro et al., 2017*) against GENCODE v23 transcripts to generate transcripts per million (TPM) abundance measures. SM-PAT-seq data, aligned and assigned as above, were converted into TPM. Data for all non-zero genes were evaluated for Spearman correlation of abundance, using the base R package (*Bates et al., 2020*).

### Estimating unique CCS reads

SM-PAT-seq data from the HEK293 and MEF time course experiments were analyzed using the program 'preseq' to model extrapolated duplication up to 1 million CCS reads based on observed duplication rates in the existing data (*Daley and Smith, 2013*).

## Cell culture

Primary MEFs were generated from E14.5 embryos by standard methods. The LARP4 KO MEFs were described (*Mattijssen et al., 2017*). Each MEF cell line was derived from a different embryo, all females. The MEFs used to generate the data in *Figure 3* were immortalized as described using SV40 Large-T antigen (*Mattijssen et al., 2017*), the experiment in *Figure 2* was performed using three independent WT- and 3 LARP4 KO MEF cell lines (N = 3, biological replicates). MEFs used for the IFNα ActD experiment in *Figure 5* were cultured following the 3T3 protocol for spontaneous immortalization (*Todaro and Green, 1963*). The experiment in *Figure 5* was performed using two independent WT- and 2 LARP4 KO MEF cell lines (N = 2, biological replicates). The immortalized MEFs and HEK293 cells were cultured in DMEM plus Glutamax (Gibco) supplemented with 10% heat-inactivated FBS (Atlanta Biologicals). HEK293 cells are not commonly misidentified. Nonetheless DNA from these cells was authenticated by ATCC via STR (short tandem repeat) profiling. Standardized testing (ATCC) had verified that the cells were free of mycoplasma infection.

## Transfection

For SM-PAT-Seq (N = 3 biological replicates), $5.5 \times 10^5$ HEK293 cells were seeded per well in a 6-well plate. The next morning the cells were ~85% confluent, media was replaced (2 ml with 10% FBS) and transfection was performed. Per well, 2.5 µg pCMV2-FLAG, pCMV2-FLAG-LARP4-WT or pCMV2-FLAG-LARP4-CS-Tyr, representing the 1X, 3X and 11X samples respectively, was co-transfected with 100 ng eGFP plasmid using 7.5 µl Lipofectamine 2000 (Invitrogen) (plasmids as described *Mattijssen et al., 2017*). For each condition, two wells were transfected. The next day, cells from both wells were combined; 80% of the cell suspension was passed to a 10 cm plate for RNA isolation the next day and 10% added to a well in a 6-well plate for protein isolation. One day later, 48 hr. post transfection, the cells for protein were ~80% confluent, were washed with 2 ml PBS and directly lysed into RIPA buffer (Pierce) containing protease inhibitors (Roche). The cells for total RNA isolation were washed twice with 10 ml PBS per 10 cm plate. Homogenization buffer (800 µl) containing thioglycerol (Maxwell 16 LEV simplyRNA purification kit, Promega) was added to lyse the cells. The lysate was transferred to a conical tube. The Maxwell 16 LEV simplyRNA purification kit protocol was followed using 800 µl lysis buffer and four cartridges per sample (400 µl per cartridge). The total DNase-treated RNA was eluted in 50 µl H2O per cartridge.

For LARP4 overexpression and ISG induction by poly(I:C), dA:dT(80), or AT-rich DNA, HEK293 cells were seeded at $5.5 \times 10^5$ cells per well of a 6-well plate. The next morning the cells were ~85% confluent, media was replaced (2 ml with 10% FBS) and transfection was performed. Per well, 2.5 µg pCMV2 or pCMV2-LARP4-WT was co-transfected with 100 ng eGFP plasmid and 100 ng poly(I:C) (Sigma) (N = 3, biological replicates) or 500 ng dA:dT(80) (N = 1), using 7.5 µl Lipofectamine 2000. The next day, cells were passed 1:5 to 4 wells. After 24 hr, cells were washed twice with 2 ml PBS and protein and RNA was isolated as follows. For protein, one well was used; cells were lysed directly in 80 µl of RIPA (Pierce) plus protease inhibitors. For RNA, to the remaining three wells, 100 µl homogenization buffer containing thioglycerol (Maxwell 16 LEV simplyRNA purification kit, Promega) was added per well to lyse the cells. The Maxwell 16 LEV simplyRNA purification kit protocol was followed using 200 µl lysis buffer and one cartridge per sample. Total DNase-treated RNA was eluted in 60 µl H2O per cartridge.

For *Figure 4A*, lanes 17–20, amounts transfected were 1.25 µg empty pCMV2-FLAG or -FLAG-LARP4-WT, together with 100 ng eGFP plasmid and 3 µg pUC19 or 1 kb AT-rich DNA and 9 µl Lipofectamine2000 (N = 1).

## IFNα and actinomycin D time course

Two LARP4 WT and two KO MEF cell lines immortalized by the 3T3 method (*Todaro and Green, 1963*), were seeded at a density of $2.5 \times 10^5$ cells per well in 2 ml media with 10% FBS in 6-well plates; at least four wells per time point were seeded. The next day the cells were ~90% confluent.

To all wells except 'unstimulated' was added mouse IFNα (Miltenyi) to 15000 U/ml. After 4 hr ActD was added to all wells (except t = 0, and 'unstimulated') to a concentration of 5 µg/ml. Total RNA was isolated at indicated time points. Cells were washed with 2 ml PBS, followed by lysis with 100 µl homogenization buffer containing thioglycerol per well (Maxwell 16 LEV simplyRNA purification kit, Promega). The lysate was transferred to an Eppendorf tube; the Maxwell 16 LEV simplyRNA purification kit protocol was followed using 200 µl lysis buffer and one cartridge per sample. Total DNase-treated RNA was eluted in 50 µl H2O per cartridge.

### Northern blotting

Northern blotting was as described (*Mattijssen et al., 2017*). Oligo probe sequences and hybridization temperatures can be found in *Supplementary file 6*.

### Western blotting

Cell lysates as described above were sonicated 3 times for 30 s on the highest setting in a Bioruptor (Diagenode). Lysates were cleared by centrifugation at 13,000 rpm for 20 min at 4°C. Supernatant was transferred to a new tube and total protein concentration determined by BCA (Pierce). Equal amounts of total protein were separated on a NuPAGE 4–12% Bis-Tris protein gel (Invitrogen) and transferred to a nitrocellulose membrane (ThermoFisher) by wet transfer. Primary antibodies were anti-FLAG (Sigma, F1804), anti-GFP (Santa Cruz, sc-8334) and anti-actin (Thermo Scientific, PA1-16890). Secondary antibodies were from LI-COR Biosciences, conjugated to either IRDye 800CW or 680RD. Signals were scanned using the Odyssey CLx imaging system (LI-COR Biosciences).

## Acknowledgements

We thank Alan Kessler (NICHD) for helpful comments on the manuscript, as well as Yevgen Levdansky, Eugene Valkov (NCI) and Amitabh Ranjan for discussion. This work was supported by the Intramural Research Program (HD000412-31 PGD) of the *Eunice Kennedy Shriver* National Institute of Child Health and Human Development, National Institutes of Health.

## Additional information

### Funding

| Funder | Grant reference number | Author |
| --- | --- | --- |
| Eunice Kennedy Shriver National Institute of Child Health and Human Development | HD000412-31 PGD | Richard J Maraia |

The funders had no role in study design, data collection and interpretation, or the decision to submit the work for publication.

### Author contributions

Sandy Mattijssen, Conceptualization, Data curation, Formal analysis, Supervision, Validation, Investigation, Methodology, Writing - original draft, Writing - review and editing; James R Iben, Data curation, Software, Formal analysis, Investigation, Visualization, Methodology, Writing - review and editing; Tianwei Li, Investigation, Methodology; Steven L Coon, Supervision, Investigation, Methodology, Writing - review and editing; Richard J Maraia, Conceptualization, Formal analysis, Funding acquisition, Investigation, Visualization, Methodology, Writing - original draft, Project administration, Writing - review and editing

### Author ORCIDs

Richard J Maraia https://orcid.org/0000-0002-5209-0066

### Decision letter and Author response

Decision letter https://doi.org/10.7554/eLife.59186.sa1
Author response https://doi.org/10.7554/eLife.59186.sa2

## Additional files

### Supplementary files

• Supplementary file 1. HEK293 mRNA PAT data (E, WT and CS corresponds to 1X, 3X and 11X) triplicate, for *Figure 2*.

• Supplementary file 2. WT and KO MEF mRNA PAT data, for *Figure 3*.

• Supplementary file 3. 128 ISG mRNAs induced to ≥1.5 fold higher in HEK293 by 3X LARP4 relative to 1X.

• Supplementary file 4. 194 ISG mRNAs induced to ≥2 fold higher levels in MEFs by IFNα.

• Supplementary file 5. Primers and adapters used for SM-PAT-seq.

• Supplementary file 6. Oligo probe sequences and hybridization temperatures used for northern blots.

• Transparent reporting form

### Data availability

All data generated or analysed during this study are included in the manuscript and supporting files.

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
