## [Decision Letter]

Thank you for submitting your article "Single molecule poly(A) tail-seq shows LARP4 opposes deadenylation through mRNA lifespan with most impact on short tails" for consideration by *eLife*. Your article has been reviewed by two peer reviewers, and the evaluation has been overseen by a Reviewing Editor and James Manley as the Senior Editor. The following individuals involved in review of your submission have agreed to reveal their identity: Jack D Keene (Reviewer #1); Maria R Conte (Reviewer #2).

The reviewers have discussed the reviews with one another and the Reviewing Editor has drafted this decision to help you prepare a revised submission.

Summary:

The authors showed before that La-related protein 4 (LARP4) promote poly(A) tail (PAT) lengthening and stabilization of mRNAs. Here, by using a novel single-molecule SM-PAT-seq method they demonstrate that LARP4 preferentially stabilizes mRNAs with PAT of 30-75 nucleotides. They also showed that LARP4 shapes PAT profiles throughout the mRNA lifespan.

Essential revisions:

The reviewers ask for the addition of statistical data (p-values) and changes in the Discussion including its shortening. It is too long (> 5 pages). Please see below the full reviews.

Reviewer #1:

This study of LARP4 functions is consistent with previous publications of the authors and other investigators regarding its importance for binding to the polyA-binding protein and the polyA RNA. Indeed, one of the most poorly understood problems in RNA biology has been how and why the length of polyA tail varies in cells. This paper presents a novel "single molecule" approach termed "SM-PAT-seq that demonstrated LARP4's interactions with different lengths of polyA. To test the biological dynamics of this phenomenon, the authors quantified polyA length by activating a TNFa immune response and comparing the time course of wild-type versus knockout of LARP4. The authors concluded that LARP4 "shapes" polyA length that in turn determines deadenylation and mRNA decay transcriptome-wide.

1) This is an excellent set of data that are consistent with the conclusions drawn, but a few questions remain such as the statistics used. Some of the datasets that were said to been repeated two or three times do not show p-values. Thus, it would strengthen the paper to present additional statistical values where possible.

2) While this paper seems to overlap and/or confirm data shown in the authors' previous 2017 *eLife* publication, the SM-PAT-seq and the ActD time course significantly advance our understanding of regulatory mechanisms that determine the dynamics of polyA length.

3) However, the Discussion overall seems unnecessarily long (5+ pages) and even rambling issues that would be better presented in a review paper. I would suggest shortening or removing sections that may not be directly pertinent to these data and conclusions, and instead, focusing on the potential of the proposed model. As an example, in the fourth paragraph of the Discussion: I do not understand why the authors pre-empted a defensive criticism regarding the paper's scope pertaining the correlation of PAT length and mRNA decay?

Reviewer #2:

The manuscript by Mattijssen et al., reports the effect of the RNA binding protein LARP4 on mRNA deadenylation, emphasising its impact on short polyA tails. This study significantly extends a previous reported article of the same authors (Mattijssen et al., 2017).

With the proviso that I am perhaps not best placed to assess some of the experimental procedures of the work, in my view this very well executed and substantial study will significantly advance our understanding of mechanisms of translation and mRNA stability and undoubtedly deserves publication on *eLife*. It will have a great impact and will be of great interest to scientists working on La-related proteins, PABP, mRNA deadenylation and regulation of gene expression.

My only critique regards some of the Discussion. In the proposed model of LARP4-PABP-sensitive protection from critical deadenylation the authors suggest that LARP4 slows deadenylation process from 75 nt (two PABP) to 37 nt (one PABP) by interacting to polyA and PABP (and this stabilises PABP on the PAT). In other words, could the small transition peak in Figure 5E at approximately 60 nt be one PABP and one LARP4 molecule bound to polyA? I am not sure this is what the authors are suggesting and I would appreciate some clarity. If this is the case, a PAM2w mutant of LARP4A or ΔNTR should behave differently, as these mutants have a lower ability to associate with polyA (Cruz-Gallardo et al).

The other confusing point, to me, is the seventh paragraph of the subsection “A proposed model of LARP4-PABP-sensitive protection from critical deadenylation” – [By extrapolation…the point at which only 2 PABP remain on the PAT would fit with this transition point at 100-110 nt]. I am somewhat confused here by whether the authors mean the transition point at 110nt is 2 or 3 PABP – the sentence here seems to contradict previous statements.

Furthermore, to understand this mechanism in more depth, it may be interesting to elaborate on the role of PAM2 competition binding between LARP4 and Tob2 (of the Ccr4-Not-Caf1-Tob2 deadenylation complexes).

---

## [Author Response]

Essential revisions:The reviewers ask for the addition of statistical data (p-values) and changes in the Discussion including its shortening. It is too long (> 5 pages). Please see below the full reviews.Reviewer #1:This study of LARP4 functions is consistent with previous publications of the authors and other investigators regarding its importance for binding to the polyA-binding protein and the polyA RNA. Indeed, one of the most poorly understood problems in RNA biology has been how and why the length of polyA tail varies in cells. This paper presents a novel "single molecule" approach termed "SM-PAT-seq that demonstrated LARP4's interactions with different lengths of polyA. To test the biological dynamics of this phenomenon, the authors quantified polyA length by activating a TNFa immune response and comparing the time course of wild-type versus knockout of LARP4. The authors concluded that LARP4 "shapes" polyA length that in turn determines deadenylation and mRNA decay transcriptome-wide.1) This is an excellent set of data that are consistent with the conclusions drawn, but a few questions remain such as the statistics used. Some of the datasets that were said to been repeated two or three times do not show p-values. Thus, it would strengthen the paper to present additional statistical values where possible.

We added thirteen *p* values to the revised manuscript that weren’t in the original version. All of these are noted in the figure legends, and some in the figures as well, and two are mentioned in the main text. The vast majority of these *p* values are very strongly significant. These additions strongly support the data and therefore significantly strengthen multiple aspects of the paper overall. We thank the reviewer for this excellent suggestion.

2) While this paper seems to overlap and/or confirm data shown in the authors' previous 2017 eLife publication, the SM-PAT-seq and the ActD time course significantly advance our understanding of regulatory mechanisms that determine the dynamics of polyA length.

We appreciate that the reviewer recognizes SM-PAT-seq as an advance. This is indeed a new method designed to have attributes for transcriptome-wide PAT analysis. Multiple of the *p* values in the revised manuscript strengthen SM-PAT-seq data that compare different LARP4 activity states including null. However, we also added *p* values to strengthen data demonstrating the performance of the SM-PAT-seq method itself, as quantitative of transcript abundance as compared to RNA-seq, *p* value of 2.2e-16 (subsection “Transcriptome wide, long-read, single-molecule poly(A)-tail, SM-PAT-seq”, and legend of Figure 1—figure supplement 1A). And also, as quantitative of transcript abundance in triplicate compared to triplicate northern blotting for GFP mRNA; R**^2^** = 0.96, *p* = 0.000035 noted in the subsection “LARP4 expression leads to poly(A) tail net-lengthening of thousands of mRNAs” and as a new figure panel, Figure 2G. The graphic data in Figure 2G strengthens the point that protection of long PATs by LARP4 appears to confer stability to stable mRNA (GFP mRNA). Other attributes of our SM-PAT-seq method are notable. Although it reports sequence from single mRNA molecules, each CCS read is derived from multiple subreads of the same molecule. Thus a CCS read of a ~1 kb amplicon (e.g., a 3’ UTR) would correspond to ~200 Illumina reads dispersed on an mRNA, but with confidence in its single molecule derivation (Figure 1—figure supplement 1B), the latter of which is lacking from the short read technology. Finally, the method was designed to produce small circles as amplicons rather than try to obtain native mRNA 5’ ends because this improves read accuracy by increasing the number of subreads per unit time during the run (see Figure 1).

We appreciate that the reviewer recognizes that use of SM-PAT-seq with ActD advances understanding of regulatory mechanisms. We discuss the findings in the context of the model developed in response to the reviewer’s comment 3 below.

3) However, the Discussion overall seems unnecessarily long (5+ pages) and even rambling issues that would be better presented in a review paper. I would suggest shortening or removing sections that may not be directly pertinent to these data and conclusions, and instead, focusing on the potential of the proposed model. As an example, in the fourth paragraph of the Discussion: I do not understand why the authors pre-empted a defensive criticism regarding the paper's scope pertaining the correlation of PAT length and mRNA decay?

I appreciate that the Discussion would seem too long (5+ pages) and agree with the reviewer’s suggestion that focus on “the potential of the proposed model” would improve the manuscript including because it was also noted in reviewer #2 comments. We revised the Discussion accordingly and it was shortened to ~3.5 pages. We followed this reviewer’s suggestion to focus on the potential of the proposed model. The new Figure 6 panels are graphics to help describe and interpret the differences in the SM-PAT-seq profiles that reflect the presence and absence of cellular LARP4. The C panel is relevant to interpretation of the A and B panels and specific issues raised by reviewer #2. We are thankful for the suggestion to focus on the model. We believe that the model and the manuscript have been improved.

In summary, I am thankful to have received such a constructive critique.

Reviewer #2:The manuscript by Mattijssen et al., reports the effect of the RNA binding protein LARP4 on mRNA deadenylation, emphasising its impact on short polyA tails. This study significantly extends a previous reported article of the same authors (Mattijssen et al., 2017).With the proviso that I am perhaps not best placed to assess some of the experimental procedures of the work, in my view this very well executed and substantial study will significantly advance our understanding of mechanisms of translation and mRNA stability and undoubtedly deserves publication on eLife. It will have a great impact and will be of great interest to scientists working on La-related proteins, PABP, mRNA deadenylation and regulation of gene expression.

I thank the Reviewer for the positive comments, insightful and helpful critique, and clarifying questions.

After submission of the original version of the manuscript and while it was being reviewed we had the opportunity to engage in further consideration of the model that was represented in the previous Figure 5E and to consult an expert in the field. This resulted in our realization that some details should be revised. The comments and questions raised by the reviewer indeed reflect inconsistencies that were revised. We replaced the small graphic that comprised previous Figure 5E and created the new Figure 6 as an expanded working model to serve to help interpret our data. The presentation also specifically addresses the points raised by the reviewer. To accompany the graphic and also focus more on the potential of the model as suggested by reviewer #1, the Discussion was reorganized to more clearly describe it.

Panel C of the new Figure 6 schematically depicts various states of three and two PABP molecules bound to poly(A) of different lengths, in a style similar to Webster et al., 2018) to help model the different size PATs in the figure panels A and B as potential PAT-PABP complexes.

My only critique regards some of the Discussion. In the proposed model of LARP4-PABP-sensitive protection from critical deadenylation the authors suggest that LARP4 slows deadenylation process from 75 nt (two PABP) to 37 nt (one PABP) by interacting to polyA and PABP (and this stabilises PABP on the PAT). In other words, could the small transition peak in Figure 5E at approximately 60 nt be one PABP and one LARP4 molecule bound to polyA? I am not sure this is what the authors are suggesting and I would appreciate some clarity. If this is the case, a PAM2w mutant of LARP4A or ΔNTR should behave differently, as these mutants have a lower ability to associate with polyA (Cruz-Gallardo et al).

To be clear, I want to note that I believe I was mistaken to assign the ~75 nt peak in previous Figure 5E as having a number of PABPs. I have come to believe that according to current models, peaks of 75 nt in PAT profiles would likely be bound by three PABPs. In the Results section of the revised manuscript we now note “The ~75 nt peak seen in Figure 5C is consistent with mRNAs that would harbor three PABP molecules (Webster et al., 2018)”. This is also detailed in the Discussion.

With regard to clarification on the issue of whether the small transition peak at approximately 60 nt could include a LARP4 molecule, we are confident that LARP4 is undetectable in the KO cells, however, we revised the Discussion to note “It is also possible that profile differences may reflect presence/absence of proteins other than PABP.”

Finally, we agree that if LARP4 uses its NTR to stabilize the PABP-PAT complex by poly(A) binding or MLLE binding, a ΔNTR mutant should behave differently. We indeed look forward to analyze results from the suggested mutant and its controls, however this data is not presently available and the analysis is beyond the scope of this paper.

The other confusing point, to me, is the seventh paragraph of the subsection “A proposed model of LARP4-PABP-sensitive protection from critical deadenylation” – [By extrapolation…the point at which only 2 PABP remain on the PAT would fit with this transition point at 100-110 nt]. I am somewhat confused here by whether the authors mean the transition point at 110nt is 2 or 3 PABP – the sentence here seems to contradict previous statements.

I apologize for the confusion and unintentional contradiction. The revised Discussion is hopefully clear: “We suspect that the 60-75 nt peak in KO represents PATs bound by three PABPs in intermediate states of dissociation/peeling off (Webster et al., 2018) (below) that are being shortened to accumulate as the peak of ~40-50 nt PATs bound by two PABPs. By extrapolation, PATs bound by two PABP-dimers would appear in the approximate 90-110 nt range in these profiles, the region of KO and WT overlap (Figure 6A).”

Furthermore, to understand this mechanism in more depth, it may be interesting to elaborate on the role of PAM2 competition binding between LARP4 and Tob2 (of the Ccr4-Not-Caf1-Tob2 deadenylation complexes).

We appreciate the interest in elaborating on the role of PAM2 competition binding between LARP4 and Tob2 of the Ccr4-Not-Caf1-Tob2 deadenylation complexes. We expanded on this as suggested.

The last two paragraphs of the revised Discussion include:

“In this model, decreased PABP-PAT stability would be compensated by interactions with LARP4 (Cruz-Gallardo et al., 2019; Mattijssen et al., 2017; Yang et al., 2011). […] Effects on long PATs and stable mRNA demonstrated here suggest the possibility that LARP4 can influence mRNA early in its lifespan with penetrant consequence later. Thus LARP4 raises issues regarding the long and short of PAT metabolism.”

In conclusion, I thank the Reviewer for a very helpful critique.